# Inhibitory Effects of Nitrogenous Metabolites from a Marine-Derived *Streptomyces bacillaris* on Isocitrate Lyase of *Candida albicans*

**DOI:** 10.3390/md20020138

**Published:** 2022-02-13

**Authors:** Beomkoo Chung, Ji-Yeon Hwang, Sung Chul Park, Oh-Seok Kwon, Eunji Cho, Jayho Lee, Hyi-Seung Lee, Dong-Chan Oh, Jongheon Shin, Ki-Bong Oh

**Affiliations:** 1Department of Agricultural Biotechnology, College of Agriculture and Life Sciences, Seoul National University, Seoul 08826, Korea; beomkoo01@snu.ac.kr (B.C.); eunji525@snu.ac.kr (E.C.); jayho@snu.ac.kr (J.L.); 2Natural Products Research Institute, College of Pharmacy, Seoul National University, Seoul 08826, Korea; yahyah7@snu.ac.kr (J.-Y.H.); sungchulpark@snu.ac.kr (S.C.P.); ideally225@snu.ac.kr (O.-S.K.); dongchanoh@snu.ac.kr (D.-C.O.); 3Marine Natural Products Chemistry Laboratory, Korea Institute of Ocean Science and Technology, Busan 49111, Korea; hslee@kiost.ac.kr

**Keywords:** *Streptomyces bacillaris*, bacillimide, bacillapyrrole, isocitrate lyase, *Candida albicans*

## Abstract

Two nitrogenous metabolites, bacillimide (**1**) and bacillapyrrole (**2**), were isolated from the culture broth of the marine-derived actinomycete *Streptomyces bacillaris*. Based on the results of combined spectroscopic and chemical analyses, the structure of bacillimide (**1**) was determined to be a new cyclopenta[*c*]pyrrole-1,3-dione bearing a methylsulfide group, while the previously reported bacillapyrrole (**2**) was fully characterized for the first time as a pyrrole-carboxamide bearing an alkyl sulfoxide side chain. Bacillimide (**1**) and bacillapyrrole (**2**) exerted moderate (IC_50_ = 44.24 μM) and weak (IC_50_ = 190.45 μM) inhibitory effects on *Candida albicans* isocitrate lyase, respectively. Based on the growth phenotype using *icl*-deletion mutants and *icl* expression analyses, we determined that bacillimide (**1**) inhibits the transcriptional level of *icl* in *C. albicans* under C_2_-carbon-utilizing conditions.

## 1. Introduction

The glyoxylate cycle is a modified tricarboxylic acid (TCA) cycle. Some bacteria, algae, fungi, and protozoa can grow with acetate as the sole carbon source by using it to synthesize TCA cycle intermediates in the glyoxylate cycle [1]. This cycle is composed of several TCA cycle enzymatic reactions plus two additional enzymes: isocitrate lyase (ICL), which cleaves isocitrate into glyoxylate and succinate, and malate synthase, which converts glyoxylate and acetyl-CoA to malate [2]. The glyoxylate cycle allows microbial pathogens such as *Candida albicans* and *Mycobacterium tuberculosis* to catabolize fatty acids via their breakdown to acetyl-CoA when available simple sugars such as glucose do not exist in the host environment [3,4]. In particular, the virulence and persistence of a *C. albicans* mutant strain lacking the ICL enzyme were remarkably decreased in a mouse model of systemic candidiasis [5,6,7]. Moreover, ICL has been considered a promising target for the development of antimicrobial agents since the glyoxylate cycle is absent in mammalian cells.

Heterocyclic chemistry, a major division of organic chemistry, includes compounds that have at least two different elements that have ring-membered atoms in their cyclic structure [8,9]. These compounds have been a focus in both biological and industrial fields due to their wide applications in cosmetics, polymers, and therapeutic agents [10]. Pyrrole, a five-membered heterocyclic aromatic compound, has gained an immense amount of attention based on its abundant pharmaceutical effects, placing pyrrole and its analogs at the center of attention [11]. In particular, the combination of diverse pharmacophores with pyrrole ring systems has allowed the development of compounds exhibiting better activities, such as anti-HIV [12], antiproliferative [13], anticancer [14], antimicrobial [15], antiviral [16], anti-inflammatory [17] and antimalarial [18] activities.

During the course of our search for bioactive compounds from marine-derived microbes, we collected a strain of *Streptomyces bacillaris* MBTC38 from marine sediment from Jeju Island in the Republic of Korea. We identified four lactoquinomycins and determined the mode of action of lactoquinomycin A, which showed the strongest activity against pathogenic bacteria [19]. In a continuous study with the MBTC38 strain, we observed moderate inhibition by the culture extract against the *C. albicans*-derived ICL enzyme, prompting our extensive investigation. The cultivation of this strain, followed by extraction and chromatographic separation, yielded two nitrogenous metabolites. Subsequently, their structures were identified by combined spectroscopic and chemical analyses, including the electronic circular dichroism (ECD)-based computational method (Figure 1). In this study, we reported the structure of bacillimide (**1**), a cyclopenta[c]pyrrole-1,3-dione of a very rare structural class [20]. The structure of previously reported bacillapyrrole (**2**) was fully characterized for the first time in this work. Although derivatives of the two compounds exhibit diverse bioactivities, including antifungal [21], anti-inflammatory [22], and antiviral [20] properties, their inhibitory activity against ICL has not been investigated. Herein, we first evaluated the potential of two compounds as inhibitors of *C. albicans* ICL.

## 2. Results and Discussion

### 2.1. Structural Elucidation

The molecular formula of compound **1** was deduced to be C_8_H_11_NO_3_S bearing four degrees of unsaturation by negative high-resolution fast atom bombardment mass spectrometry (HR-FAB-MS) analysis ([M − H]^−^ *m*/*z* 200.0387, calcd 200.0387). The ^13^C nuclear magnetic resonance (NMR) data of this compound showed signals of two carbonyl carbons at δ_C_ 180.6 and 177.9. A strong absorption band at 1634 cm^−1^ in the infrared (IR) data indicated these to be amide carbonyls. Then, the presence of a single nitrogen inherent in the MS data linked these carbonyls to an imide group. With the aid of heteronuclear single quantum coherence (HSQC) NMR data, the combination of the remaining carbons and their attached protons diagnosed a nonprotonated carbon (δ_C_ 57.0), two methines including an oxymethine (δ_C_/δ_H_ 62.4/2.85, 74.2/4.26), two methylenes (δ_C_/δ_H_ 33.7/2.18 and 2.07, 33.6/1.76 and 1.50), and a methyl group (δ_C_/δ_H_ 12.5/2.14) (Table 1). Additionally, two exchangeable protons (δ_H_ 11.49, 5.28) were found in the ^1^H NMR data. The lack of sp^2^ carbons except for two carbonyls, in conjunction with the four unsaturation degrees inherent in the MS data, suggested **1** is a bicyclic compound.

The planar structure of **1** was determined by a combination of ^1^H correlation spectroscopy (COSY) and heteronuclear multiple bond correlation (HMBC) experiments. First, a linear array of two methylenes (C-4 and C-5) with oxymethine (C-6) was revealed by the COSY correlations of the attached protons (H_2_-4-H_2_-5-H-6) as well as the mutual HMBC correlations among these protons (Figure 2). The oxygenated functionality at C-6 was defined as a hydroxyl group by the COSY correlation of H-6 (δ_H_ 4.26) with an exchangeable proton (δ_H_ 5.28) as well as the key HMBC correlations of the latter proton with neighboring carbons: OH-6/C-5 and C-6. Due to the lack of significant proton-proton correlations, the subsequent extension of the C-4-C-6 partial structure was accomplished with the HMBC data. That is, the nonprotonated carbon (C, δ_C_ 57.0) and a methine group (CH, δ_C_/δ_H_ 62.4/2.85) were placed at the neighboring C-3 and C-7 positions based on several of their HMBC correlations with the C-4-C-6 group: H_2_-4/C and CH, H_2_-5/C and CH, H-6/C, and H (δ_H_ 2.85)/C-4 and C-5. Distinguishing between C-3 and C-7 was also achieved by the crucial three-bond correlation of the methine carbon with the 6-OH proton, placing this methine at C-7 and the remaining nonprotonated carbon at C-3. The weak vicinal coupling (*J* = 1.0 Hz) between H-6 and H-7 was attributed to the 5-membered ring and attachment of the electron-withdrawing 6-OH and is discussed later. Further examination of the HMBC data directly linked C-3 and C-7, constructing a cyclopentane moiety (C-3-C-7) by the correlation at H-7/C-3. In addition, the distinct chemical shifts (δ_C_/δ_H_ 12.5/2.14) of an isolated methyl group (C-9) were indicative of a methylsulfide group that was directly linked at C-3 by the three-bond HMBC correlation at H_3_-9/C-3.

Compound **1** was structurally defined as a bicyclic molecule by HR-FAB-MS analysis. Since a cyclopentane moiety was unveiled by combined 2-D NMR analyses, the remaining moiety must be composed of the predescribed imide group attached at the C-3 and C-7 open ends of the cyclopentane, forming a 5-membered succinimide (pyrrole-2,5-dione) moiety. This interpretation was confirmed by the HMBC data in which both carbonyl carbons (δ_C_ 179.4 and 176.4) showed long-range correlations with the H-7 methine proton (Figure 2). An additional correlation with H_2_-4 assigned the former carbon at C-2, leaving the latter one at C-8. Thus, the structure of **1**, designated bacillimide, was determined to be a cyclopenta[c]pyrrole-1,3-dione bearing a methylsulfide group. A literature study showed that **1** belongs to a very rare bacterial metabolite group preceded only by the recently reported nitrosporeusines from the Arctic Streptomyces nitrosporeus [20]. The most remarkable structural difference was the methylsulfide group of **1** replacing the p-hydroxy-benzenecarbothio group of nitrosporeusines. Prompted by the antiviral activity of nitrosporeusines, several derivatives have been designed and synthesized, resulting in the identification of significant anti-inflammatory agents [23].

Bacillimide (**1**) possesses three stereogenic centers at C-3, C-6, and C-7, which are identical to nitrosporeusines. Based upon the results of combined CD and crystallographic analyses, nitrosporeusines A and B were assigned the 3S, 6S, and 7R and 3R, 6S, and 7S configurations, respectively, having opposite ring junctures [20]. In our initial nuclear overhauser effect (NOESY) analysis, a conspicuous cross-peak at H-7/H_3_-9 assigned a cis ring juncture as nitrosporeusines (Figure 3). Another cross-peak was found at 6-OH/H-7, suggesting spatial proximity between these groups. This interpretation was supported by the chemical shifts and splitting patterns of several protons of **1** in the ^1^H NMR data in both DMSO-*d_6_* and MeOH-*d_4_*_,_ showing good accordance with those of synthetic analogs (Appendix A) [23]. However, the NOESY correlation of an exchangeable proton requires more decisive evidence. This problem was solved by chemical derivatization of **1** to a more spatially occupying analog bearing nonexchangeable protons at the stereogenic centers. Thus, treatment of **1** with benzoic anhydride produced benzoyl bacillimide (**1a**). After the ^13^C and ^1^H NMR assignments, the NOESY data of **1a** showed spatial proximity between H_3_-9 and H-2′, unambiguously confirming the 3*R**, 6*S**, and 7*S** relative configurations.

The absolute configurations of **1** were initially investigated by the α-methoxy α-trifluoromethylphenylacetic acid (MTPA) method. Possibly due to severe spatial crowding, however, the production of MTPA esters was partially successful (compound **1** was highly reactive with (*R*)-MTPA-Cl, readily producing the (*S*)-MTPA ester). However, it was inert against (*S*)-MTPA-Cl under diverse reaction conditions. Consequently, the absolute configurations were determined by the ECD-based computational method. As shown in Figure 4, the measured CD profile of **1** was highly compatible with the calculated ECD of 3*R*, 6*S*, and 7*S* configurations, the same as nitrosporeusine B. A similar analysis against benzoyl derivative **1a** also showed identical results supporting the ECD-based absolute configurations of **1**. Thus, the structure of bacillimide (**1**) was determined to be a cyclopenta[c]pyrrole-1,3-dione bearing a methylsulfide group.

The molecular formula of compound **2** was established as C_8_H_1__2_N_2_O_2_S by positive high-resolution electrospray ionization mass spectrometry (HR-ESI-MS) analysis ([M + H]^+^ *m*/*z* 201.0687, calcd C_8_H_13_N_2_O_2_S, 201.0692). The ^13^C, ^1^H and HSQC NMR data of this compound diagnosed signals to a carbonyl carbon (δ_C_ 160.8), four aromatic/olefinic (δ_C_/δ_H_ 126.0, 121.4/6.83, 110.0/6.74, and 108.6/6.07), two methylene (δ_C_/δ_H_ 53.6/3.00 and 2.86, and 32.6/3.60 and 3.52), and a methyl group (δ_C_/δ_H_ 38.1/2.58). Additionally, two exchangeable protons (δ_H_ 11.45 and 8.26) were found in these data (Table 1). Further examination of spectroscopic data was informative of key structural motifs. That is, for the carbonyl functionality, a strong absorption band at 1682 cm^−1^ in the IR data was indicative of an amide group. The small proton-proton coupling constants (*J* 2.0–1.5 Hz) among the aromatic/olefinic protons were interpreted as being derived from a pyrrole-type moiety [24]. Finally, the remarkably deshielded chemical shifts of the isolated methyl group were determined to be sulfur-bearing, which was thought to be attributable to a sulfinylmethyl group based on a strong absorption band at 1039 cm^−1^ in the IR data. According to this information, the SciFinder study revealed that compound **2** was indeed a commercially available compound (CAS #1928723-80-1). However, spectroscopic data were unreported, urging full chemical characterization of this compound.

Combined ^1^H COSY and HMBC analyses of **2** readily defined the aromatic moiety as a 2-substituted pyrrole (1-NH, C-2-C-5) (Figure 2). However, substitution at this pyrrole moiety was unmade by the lack of proton-carbon correlations in the given HMBC experiments (J_CH_ = 2 and 8 Hz). In addition, the COSY data revealed a linear array of an exchangeable proton (7-NH, *δ*_H_ 8.26) and two methylenes (8-CH_2_ and 9-CH_2_) that were supported by the HMBC correlations at 7-NH/C-8, H_2_-8/C-9, and H_2_-9/C-8. Similarly, the methylsulfoxide group was placed at C-10 by the HMBC correlations at H_2_-9/C-10 and H_3_-10/C-8 and C-9. The amide group was also confirmed by the HMBC correlations at 7-NH/C-6 and H_2_-8/C-6. Although it was not directly proven by spectroscopic data, two open ends of 2-D NMR correlations at C-5 of pyrrole and the C-6 carbonyl carbon were rationally linked to each other. Thus, the structure of compound **2**, designated bacillapyrrole, was fully characterized as N-(2-(methylsulfinyl)ethyl)-1H-pyrrole-2-carboxamide.

### 2.2. ICL Inhibitory Activity and Antifungal Activity of Isolated Compounds

Two isolated compounds (**1** and **2**) were tested for inhibitory activity against *C. albicans* ICL based on previously reported methods [25]. The half-maximal inhibitory concentration (IC_50_) values of the two compounds and 3-nitropropionate, known as a potent ICL inhibitor, are shown in Table 2. Compounds **1** and **2** exhibited moderate and weak inhibitory activities toward the ICL enzyme of *C. albicans*, with IC_50_ values of 44.24 µM and 190.45 µM, respectively, which were less than that of a known ICL inhibitor, 3-nitropropionate (IC_50_ = 21.49 µM). To confirm the type of inhibition, kinetics analysis was carried out with **1** at the IC_50_ or twofold IC_50_ based on a Lineweaver and Burk plot (Figure 5). The inhibitor constant was obtained by a Dixon plot. Bacillimide (**1**) behaved as a mixed inhibitor, with an inhibitor constant (*K*i) value of 0.42 mM. Fungal growth inhibition tests showed that compounds **1** and **2** did not exert inhibitory effects on ATCC10231 cultured in glucose (Table 2).

### 2.3. Inhibition of C_2_ Carbon Source Utilization

It was expected that ICL inhibitory compounds would suppress nutrient uptake from C_2_ carbon sources and disturb the survival of pathogens in macrophages. To evaluate whether bacillimide (**1**) affects the usage of C_2_ substrate, five *C. albicans* strains (ATCC10231, ATCC10259, ATCC11006, ATCC18804, and SC5314) were grown in yeast nitrogen base (YNB) broth containing either 2% glucose or 2% potassium acetate as the sole carbon source. Bacillimide (**1**) inhibited the growth of *C. albicans* in acetate-containing broth at a concentration of 256 µg/mL but had no inhibitory effect on fungal cells grown in glucose (Table 3). These results demonstrated that bacillimide (**1**) inhibits ICL-mediated proliferation of the fungus under C_2_-carbon-utilizing conditions.

### 2.4. Effects of Bacillimide (***1***) on Growth Phenotype and icl Gene Expression

To confirm whether bacillimide (**1**) affects the cell phenotype of the wild-type and the *icl*-deletion mutant under C_2_-assimilating conditions, an in vitro growth assay was carried out using *C. albicans* SC5314 (wild-type), *icl*-deletion mutant (MRC10), and *icl*-complementary mutant (MRC11). These strains were streaked onto YNB agar plates supplemented with 2% glucose or 2% potassium acetate with or without 256 µg/mL **1**. All strains normally developed their phenotypes on both YNB agar plates with glucose and glucose plus **1**. However, the MRC10 strain did not grow on an agar medium in which acetate was the sole carbon source. Moreover, no cell growth was observed on the YNB agar plate with acetate plus **1** (Figure 6a). These results indicated that the ICL enzyme is related to the growth of *C. albicans* on the C_2_ substrate.

We further conducted semiquantitative reverse-transcription (RT)-PCR to confirm the effects of bacillimide (**1**) on the mRNA expression of *icl*. The *icl*-specific PCR product was not detected in the whole strains cultured in YNB broth with glucose, while *icl* expression was strongly induced when strains were cultured in YNB broth containing acetate due to activation of the glyoxylate cycle. The intensity of the PCR band corresponding to the mRNA expression of *icl* was reduced with increasing concentrations of **1** in the cell cultures (Figure 6b). The expression of GAPDH, a housekeeping gene in *C. albicans*, was uniformly observed in all cell cultures regardless of the presence of **1**. These results indicate that bacillimide (**1**) inhibits *icl* expression in *C. albicans* under C_2_-carbon-utilizing conditions.

Based on the overall experimental results, bacillimide (**1**) diminished both the ICL enzyme activity and transcriptional level of *icl*. Bacillimide (**1**) inhibited ICL enzyme activity at a concentration of 256 µg/mL in the growth phenotype assay, while it initiated a decrease in the mRNA expression of *icl* at a concentration of 64 µg/mL. Moreover, 128 µg/mL **1** impeded almost all *icl* gene expression (Table 3 and Figure 6b). Hence, we deduced that transcriptional inhibition would be a preferential target of bacillimide (**1**). Further studies are required to clarify the relationship between the reduction in *icl* expression and inhibition of ICL enzyme activity and to identify the main cellular target of bacillimide (**1**).

## 3. Materials and Methods

### 3.1. General Experimental Procedure

Optical rotations were measured on a JASCO P1020 polarimeter (Jasco, Tokyo, Japan) using a 1 cm cell. Ultraviolet (UV) spectra were acquired with a Hitachi U-3010 spectrophotometer (Hitachi High-Technologies, Tokyo, Japan). ECD spectra were recorded on a Chirascan plus CD spectrometer (Applied Photophysics Ltd., Leatherhead, Surrey, UK). IR spectra were recorded on a JASCO 4200 FT-IR spectrometer (Jasco, Tokyo, Japan) using a ZnSe cell. ^1^H and ^13^C NMR spectra were measured in DMSO-*d_6_* or MeOH-*d_4_* solutions on a Bruker Avance −400, −500, −600, or −800 instrument (Billerica, MA, USA), with solvent peaks at δ_H_ 2.50/δ_C_ 39.50 and δ_H_ 3.31/δ_C_ 49.00 as their internal standards. High-resolution ESI mass spectrometric data were obtained at the National Instrumentation Center for Environmental Management (Seoul, Korea) and were acquired using an AB Sciex 5600 QTOF HR-MS instrument (Sciex, MA, USA). High-performance liquid chromatography (HPLC) analysis was conducted using a Shimadzu SCL-10A (Shimadzu, Tokyo, Japan) control system connected to a UV–Vis SPD-10A detector (Shimadzu). All solvents used were of spectroscopic grade or distilled from glass prior to use.

### 3.2. Taxonomic Identification and Fermentation

In our previous study, the isolated bacterial strain MBTC38 from underwater sediment on Jeju Island showed 100% similarity with *Streptomyces bacillaris,* and we assigned the strain *Streptomyces bacillaris* MBTC38 (GenBank accession number: MK402083.1) [19]. The *Streptomyces bacillaris* strain MBTC38 was sporulated on colloidal chitin agar plates (4 g of chitin, 0.7 g of K_2_HPO_4_, 0.5 g of MgSO_4_·7H_2_O, 0.3 g of KH_2_PO_4_, 0.01 g of FeSO_4_·7H_2_O, 0.001 g of MnCl_2_·4H_2_O, 0.001 g of ZnSO_4_·7H_2_O, 20 g of agar, and 17 g of sea salt in 1 L of distilled water) at 28 °C for 10 days. Mature spores were inoculated into 500 mL of colloidal chitin liquid medium and incubated at 28 °C for 7 days on a rotatory shaker.

### 3.3. Extraction and Isolation

The entire culture (160 L) was filtered through filter paper and extracted with an equal volume of EtOAc twice. The organic solvents were evaporated to dryness under reduced pressure to obtain 4.4 g of total extract. Based on inhibitory activities toward the ICL enzyme, the entire extract (4.4 g) was separated through reversed-phase HPLC (Agilent Eclipse XDB-C18, 5 µm, 9.4 × 250 mm; H_2_O-MeCN, 88:12, 2.0 mL/min) with 0.1% trifluoroacetic acid (UV detection at 254 nm) to yield a mixture of Compounds **1** and **2** as a single peak (*t*_R_ = 10.6 min). Final purification was accomplished by analytical HPLC (YMC-ODS-A column, 5 µm, 4.6 × 250 mm; H_2_O-MeOH, 75:25, 0.7 mL/min; **1**: *t*_R_ = 15.4 min, 9.1 mg, **2**: *t*_R_ = 14.2 min; 7.8 mg).

#### 3.3.1. Bacillimide (**1**)

Brown oil; [α]$\begin{array}{c} 25\\ D \end{array}$ +14 (c 0.1, MeOH); UV (MeOH) λ_max_ (log ε) 213 (2.66), 256 (3.19) nm; ECD (*c* 1.0 mM, MeOH) λ_max_ (Δε) 210 (2.78), 230 (−25.57), 248 (2.50), 269 (44.23) nm; IR (ZnSe) ν_max_ 2925, 2359, 1634, 1366, 1183 cm^−1^; ^1^H and ^13^C NMR data, Table 1; HRESIMS *m*/*z* 200.0387 [M − H]^−^ (calcd for C_8_H_10_NO_3_S, 200.0387).

#### 3.3.2. Bacillapyrrole (**2**)

Light brown gum; [α]$\begin{array}{c} 25\\ D \end{array}$ +40 (*c* 0.2, MeOH); UV (MeOH) λ_max_ (log ε) 209 (1.83), 265 (2.98) nm; IR (ZnSe) ν_max_ 3282 (br), 2925, 2358, 1682, 1566, 1337, 1205, 1039 cm^−1^; ^1^H and ^13^C NMR data, Table 1; HRESIMS *m*/*z* 201.0687 [M + H]^+^ (calcd for C_8_H_13_N_2_O_2_S, 201.0692).

### 3.4. Benzoylation of Compound ***1***

Benzoylation was performed for the assignment of absolute configurations of Compound **1**. Compound **1** (0.7 mg, 3.5 μmol) in pyridine (200 μL) was added dropwise to a solution of benzoic anhydride (2.2 mg, 9.7 μmol) and dimethylaminopyridine (DMAP, 0.1 mg, 0.81 μmol) in pyridine (200 μL). The mixture was stirred at room temperature for 8 h. The reaction was diluted with EtOAc and slowly added to 1 N HCl until the pH became acidic. The crude reaction mixture was concentrated in vacuo and then purified by analytical HPLC (YMC-ODS column, 4.6 × 250 mm; 0.7 mL/min; H_2_O-MeCN gradient from 80:20 to 10:90 in 40 min) to yield benzoyl bacillimide (**1a**) (0.9 mg, 2.9 μmol, 85% yield).

#### Benzoyl Bacillimide (**1a**)

^1^H NMR (DMSO-*d*_6_, 800 MHz) δ_H_ 7.98 (2H, d, *J* = 8.0 Hz, H-2′, H-6′), 7.51 (1H, t, *J* = 7.5 Hz, H-4′), 7.55 (2H, t, *J* = 7.5 Hz, H-3′, H-5′), 5.46 (1H, d, *J* = 3.8 Hz, H-6), 3.28 (1H, overlap, H-7), 2.31 (1H, dd, *J* = 13.1, 7.1 Hz, H-4), 2.13 (3H, s, H-9), 2.10 (1H, m, H-4), 2.02 (1H, *J* = 14.3, 6.9 Hz, H-5), 1.78 (1H, m, H-5); ^13^C NMR (DMSO-*d*_6_, 800 MHz) δ_c_ Undetected (C-2, C-8), 164.8 (C, C-7′), 133.5 (CH, C-4′), 129.6 (C, C-1′), 129.2 (CH, C-2′, C-6′), 128.8 (CH, C-3′, C-5′), 78.0 (CH, C-6), 60.2 (CH, C-7), 57.9 (C, C-3), 34.3 (CH_2_, C-5), 31.2 (CH_2_, C-4), 12.6 (CH_3_, C-9); ECD (*c* 1.0 mM, MeOH) λ_max_ (Δε) 213 (3.58), 234 (0.25), 262 (37.77) nm; HRESIMS *m*/*z* 328.0612 [M + Na]^+^ (calcd for C_15_H_15_NO_4_SNa, 328.0614).

### 3.5. Electronic Circular Dichroism (ECD) Calculations

Based on density functional theory (DFT) calculations, the geometries were optimized to the ground-state energy level by Turbomole computer software. The basis parameter sets, def-SVP, and the B3-LYP functional for all atoms were employed. The calculated ECD data were measured based on the optimized structures obtained with TDDFT at the B3-LYP functional. The ECD spectra were obtained by overlapping Gaussian functions for each transition, where *σ* is the width of the band at height 1/*e*. Values ∆*E_i_* and *R_i_* represent the excitation energy and rotatory strength for transition *i*, respectively. In this work, the value of σ was 0.10 eV.
$$\Delta \varepsilon \left(E\right)=\frac{1}{2.297\times 10^{-39}}\frac{1}{\sqrt{2\pi \sigma }}\sum \left(\begin{array}{c} A\\ i \end{array}\right)\Delta E_{i}R_{i}e^{\left[-{\left(E-\Delta E_{i}\right)}^{2}/{\left(2\sigma \right)}^{2}\right]}$$

### 3.6. ICL Inhibitory Activity Assay

Expression and purification of recombinant ICL protein from *C. albicans* were performed using the following procedures with minor modifications followed by evaluation of the ICL inhibitory activity of isolated compounds [25]. The ICL inhibitory activity assay was carried out according to previously described methods [26,27]. Compounds **1**, **2**, and 3-nitropropionate, known as an ICL inhibitor, were dissolved in DMSO at 12.8 mg/mL. A total of 772 (first cuvette) and 390 (other cuvettes) µL of reaction MOP buffer containing 3.75 mM MgCl_2_, 1.27 mM *threo*-DL (+) isocitrate and 4.1 mM phenylhydrazine was mixed with 8 µL of tested samples and serially diluted (final DMSO concentration, 1%). Then, 10 µL of purified ICL (concentration = 2.5 µg/mL) was added and incubated at 37 °C for 30 min. The absorbance at 324 nm was measured by a UV mini 1240 spectrophotometer (Shimadzu, Kyoto, Japan) at 0 min and 30 min after incubation. The ICL inhibitory effect of each compound was calculated as an absorbance percentage relative to that of the sample treated with only DMSO. The half-maximal inhibitory concentration (IC_50)_ was measured by nonlinear regression analysis (GraphPad ver. 8.0, Prism). 3-Nitropropionate was used as a positive control [28].

### 3.7. In Vitro Growth Assay

An in vitro growth assay was performed under the following procedures with minor modifications [29]. Five wild-type *C. albicans strains* (SC5314, ATCC10231, ATCC10259, ATCC11006, and ATCC18804) were cultured overnight at 28 °C in YNB medium with 2% glucose and 2% potassium acetate and diluted to match the turbidity of a 0.5 McFarland standard at 530 nm wavelength. Stock solutions of **1** and amphotericin B were prepared in DMSO at 25.6 mg/mL. Each stock solution was diluted in YNB media to concentrations ranging from 0.25 to 256 µg/mL. The final DMSO concentration was maintained at 1% by adding DMSO to the medium according to CLSI guidelines. In each well of a 96-well plate, 90 µL of YNB media containing two compounds was mixed with 10 µL of broth with the test strain *C. albicans* (final concentration: 2.5 × 10^3^ cfu/mL). The plates were incubated for 2 days at 28 °C, and amphotericin B was used as a positive control. The MIC value was defined as the lowest concentration of the test compound that prevented cell growth.

### 3.8. Grow Phenotype and icl Expression Analysis

*C. albicans* SC5314 (wild-type strain), MRC10 (Δ*icl*) (*icl*-deletion strain), and MRC11 (Δ*icl* + ICL) (*icl*-complementary strain) were used to confirm the growth phenotype [7]. Three strains were cultured in YNB broth containing 2% glucose and 2% potassium acetate at 28 °C for 24 h. Cells were centrifuged (8000× *g*, 2 min) and washed with distilled water. The harvested cells were streaked on YNB agar plates containing 2% glucose or 2% potassium acetate containing 0 µg/mL or 256 µg/mL **1** and incubated at 28 °C for 2 days.

For *icl* expression analysis, overnight cultured *C. albicans* SC5314 was diluted with YNB media containing 2% glucose or 2% potassium acetate and incubated to mid-log phase. Various concentrations of **1** (64, 128, or 256 µg/mL) were added to media containing 2% potassium acetate and incubated at 28 °C for 4 h. Total RNA extraction was carried out using easy-BLUE^TM^ reagent (Intron Biotechnology), and cDNA was reverse-transcribed with a SuperScript III cDNA synthesis kit (Enzynomics). Semiquantitative RT–PCR was performed using *icl*-specific primers: 5′-ATGCCTTA CACTCCTATTGACATTCAAAA-3′ (forward), 5′-TAGATTCAGCTTCAGCCATCAA AGC-3′ (reverse), under the following conditions: initial denaturation at 95 °C for 5 min, 30 cycles of denaturation at 95 °C for 20 s, annealing at 55 °C for 30 s and elongation at 72 °C for 1 min and final extension at 72 °C for 5 min. The housekeeping gene glyceraldehyde-3-phosphate dehydrogenase (*GAPDH*) was used as a loading control.

## 4. Conclusions

Two nitrogenous metabolites, bacillimide (**1**) and bacillapyrrole (**2**), were isolated from the culture broth of the marine-derived actinomycete *Streptomyces bacillaris*. Based on combined spectroscopic and chemical analyses, the structure of **1** was determined to be a cyclopenta[*c*]pyrrole-1,3-dione bearing a methylsulfide group, thus belonging to a very rare structural class. The previously reported **2** was fully characterized as a pyrrole-carboxamide bearing an alkyl sulfoxide side chain by this work. Moreover, the absolute configuration of **1** was determined by analyses of NOESY correlation and ECD-based computational method. Bacillimide (**1**) exhibited a remarkable difference in that it was a new compound that substituted the methylsulfide group at C-3 for the *p*-hydroxy-benzenecarbothio group among similar compounds, nitrosporeusines. Because antiviral activities were observed in a previous study of nitrosporeusines, therefore, we confirmed antiviral activities of bacillimide (**1**) against coronavirus, but meaningful results were not elicited (data not shown) [20]. Alternatively, bacillimide (**1**) and bacillapyrrole (**2**) exhibited biological activities toward *C. albicans* ICL with an IC_50_ value of 44.24 µM and IC_50_ = 190.45 μM, respectively. Pathogenic strain *C. albicans* demands a glyoxylate cycle to retain virulence and persistence since there are plentiful C_2_ carbon sources, including fatty acids and acetate, in macrophages. Consequently, the strain accelerates the switch of the metabolic pathway from glycolysis to the glyoxylate cycle to utilize the C_2_ substrate when *C. albicans* is phagocytosed by macrophages [30]. The growth assay revealed that bacillimide (**1**) specifically inhibits the ICL enzyme of the glyoxylate cycle and blocks nutrient uptake from the C_2_ carbon source because no growth of *C. albicans* SC5314 (wild-type) and MRC11 (∆*icl* + ICL) was observed on the YNB agar plates containing acetate plus the compound at 256 µg/mL. Moreover, *icl* transcriptional levels reduced as a result of treatment with **1**, indicating that compound **1** inhibits *icl* mRNA expression rather than having a direct effect on the enzyme active site. Further studies are required to clarify the relationship between ICL activity inhibition and the reduction of *icl* expression and to identify the main cellular target of bacillimide (**1**).

## Figures and Tables

**Figure 1 marinedrugs-20-00138-f001:**
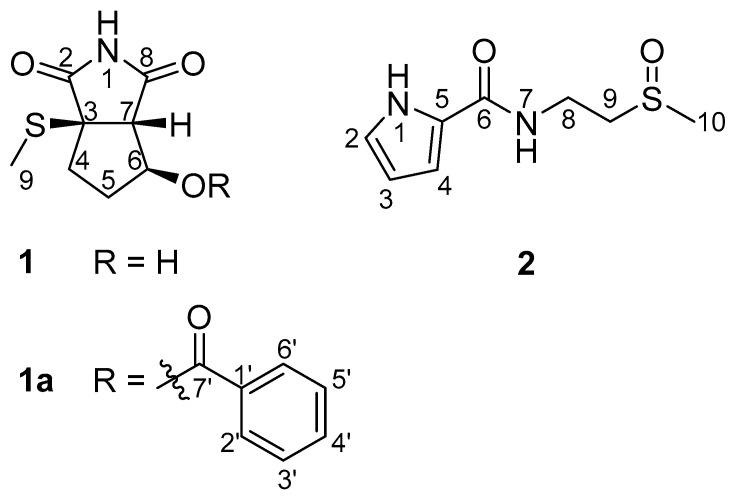
The structures of bacillimide (**1**), benzoyl bacillimide (**1a**) and bacillapyrrole (**2**).

**Figure 2 marinedrugs-20-00138-f002:**
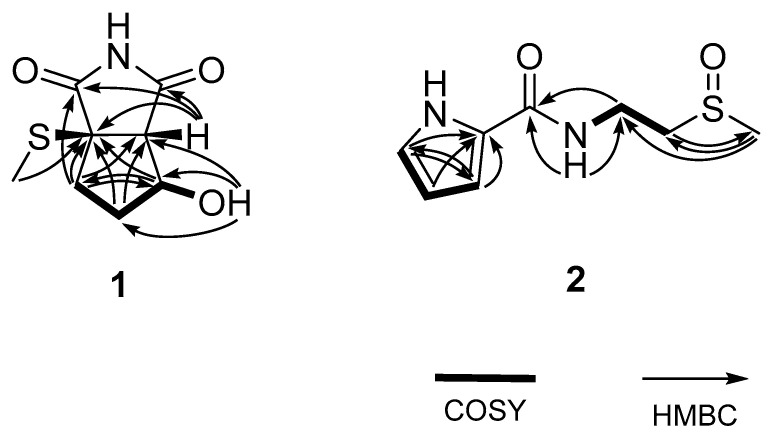
Key correlations of COSY (bold) and HMBC (arrows) experiments for compounds **1** and **2**.

**Figure 3 marinedrugs-20-00138-f003:**
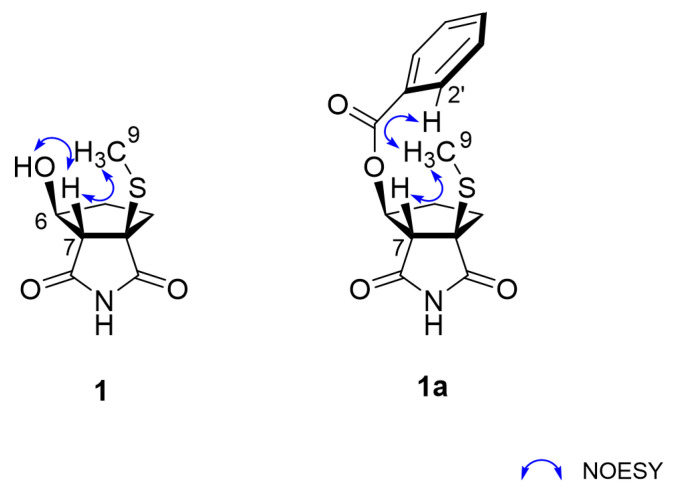
Key correlations of NOESY (arrow) experiments for compounds **1** and **1a**.

**Figure 4 marinedrugs-20-00138-f004:**
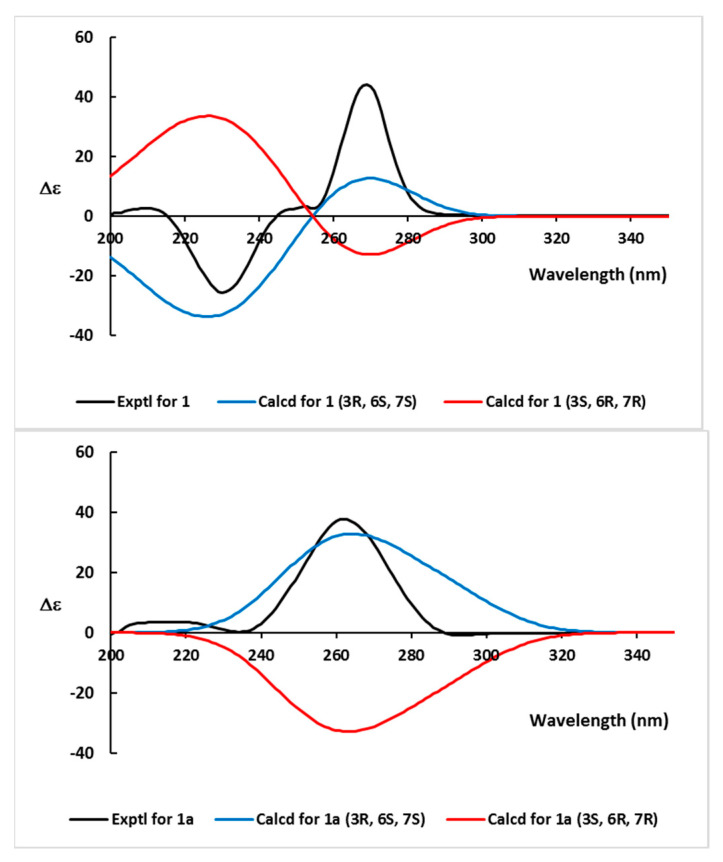
Experimental CD and ECD calculations of bacillimide (**1**) and benzoyl bacillimide (**1a**).

**Figure 5 marinedrugs-20-00138-f005:**
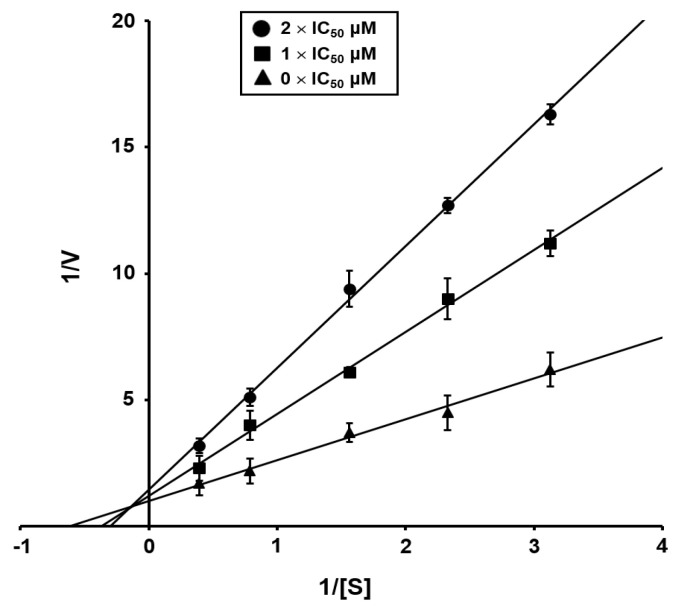
Inhibitor type of bacillimide (**1**). Lineweaver–Burk plot of ICL inhibition by bacillimide (**1**). [S] and V represent the substrate concentration (mM) and reaction velocity (ΔAbsorbace_324nm_/sec), respectively. Each data point shows the mean of three independent experiments.

**Figure 6 marinedrugs-20-00138-f006:**
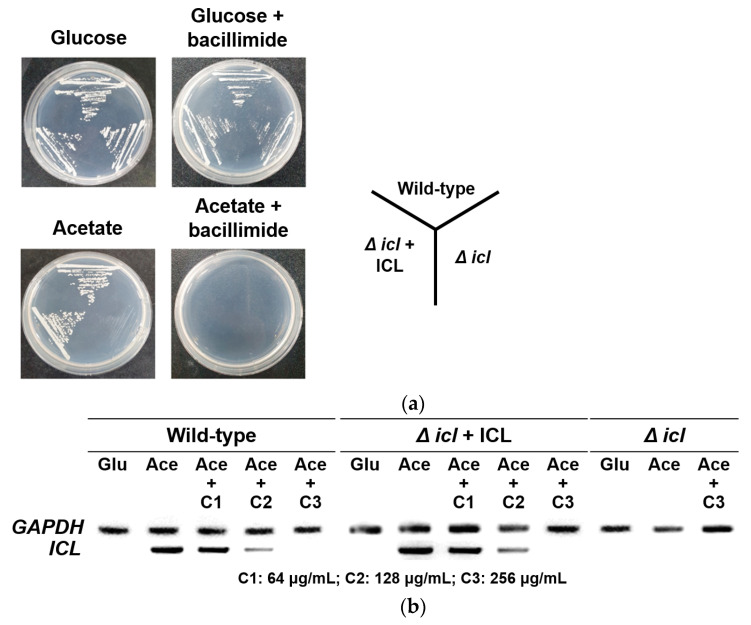
Analysis of growth phenotypes and *icl* expression in presence of bacillimide (**1**). (**a**) *C. albicans* SC5314 (wild-type strain), MRC10 (Δ*icl*) (*icl*-deletion strain) and MRC11 (Δ*icl* + ICL) (*icl*-complementary strain) were cultured on YNB agar plates with 2% glucose or 2% potassium acetate containing 0 µg/mL or 256 µg/mL bacillimide (**1**) for 2 days at 28 °C. (**b**) Mid-log phase of three *C. albicans* strains in YNB broth containing 2% glucose (Glu) were untreated, while those in YNB broth containing 2% potassium acetate (Ace) were treated with various concentrations of bacillimide (**1**) (C1: 64 µg/mL, C2: 128 µg/mL, C3: 256 µg/mL) and incubated for 4 h at 28 °C. Total RNA was extracted, and *icl* expression was analyzed with semiquantitative RT–PCR. *GAPDH*, a housekeeping gene, was used as a positive control.

**Table 1 marinedrugs-20-00138-t001:** ^1^H and ^13^C NMR data of compounds **1** and **2** in DMSO-*d*_6_ (*δ*_H_ and *δ*_C_ in ppm).

No.	1 ^a^		2 ^b^	
	*δ*_C_, Type	*δ*_H_ (*J* in Hz)	*δ*_C_, Type	*δ*_H_ (*J* in Hz)
2	179.4, C		121.4, CH	6.83, d (1.5)
3	57.0, C		108.6, CH	6.07, t (2.0)
4	33.7, CH_2_	2.18, ddd (13.0, 6.5, 1.5)	110.0, CH	6.74, d (1.5)
		2.07, dd (13.0, 6.0)		
5	33.6, CH_2_	1.76, ddd (13.5, 6.5, 1.5)	126.0, C	
		1.50, ddd (13.0, 7.0, 2.5)		
6	74.2, CH	4.26, br s	160.8, C	
7	62.4, CH	2.85, s		
8	176.4, C		32.6, CH_2_	3.60, ddd (14.0, 6.0, 1.5)
				3.52, ddd (14.0, 7.0, 1.5)
9	12.5, CH_3_	2.14, s	53.6, CH_2_	3.00, ddd (13.0, 6.5, 1.0)
				2.86, ddd (13.0, 6.0, 1.0)
10			38.1, CH_3_	2.58, s
1-NH		11.49, br s		11.45, br s
6-OH		5.28, s		
7-NH				8.26, t (4.5)

^a 1^H and ^13^C NMR data were recorded at 800 and 200 MHz, respectively. ^b 1^H and ^13^C NMR data were recorded at 600 and 150 MHz, respectively.

**Table 2 marinedrugs-20-00138-t002:** Inhibitory activity of isolated compounds against the ICL enzyme and growth of *C. albicans* ATCC10231.

Compound	ICL IC_50_ (μM)	MIC (μM) in Glucose
Bacillimide (**1**)	44.24 ± 1.05	>1273.64
Bacillapyrrole (**2**)	190.45 ± 3.86	>1280.00
3-Nitropropionate	21.49 ± 0.97	>2149.46
Amphotericin B	ND	0.5

3-Nitropropionate and amphotericin B were used as a standard inhibitor of ICL and a representative antifungal drug, respectively. ND: not determined.

**Table 3 marinedrugs-20-00138-t003:** Inhibitory effect of bacillimide (**1**) on five *C. albicans* strains grown in glucose and acetate as sole carbon sources.

Strain	MIC (μg/mL)			
	Glucose		Acetate	
	Bacillimide (1)	Amph B ^1^	Bacillimide (1)	Amph B ^1^
SC5314	>1024	0.5	256	0.5
ATCC10231	>1024	0.5	256	0.5
ATCC10259	>1024	0.5	256	0.5
ATCC11006	>1024	1	256	0.5
ATCC18804	>1024	1	256	1

*C. albicans* (2.5 × 10^3^ cfu/mL) were incubated with various concentrations of bacillimide (**1**) for 2 days at 28 °C in YNB broth containing 2% glucose and 2% potassium acetate. ^1^ Amphotericin B (Amph B) was used as a standard antifungal drug.

## Data Availability

All data is contained within this article and Appendix A.

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
