# Peer review of "Inhibitory Effects of Nitrogenous Metabolites from a Marine-Derived Streptomyces bacillaris on Isocitrate Lyase of Candida albicans"

_marinedrugs, 2022, doi:10.3390/md20020138_

Round 1

Reviewer 1 Report

In the current manuscript, the authors report on the isolation, characterization, and bioactivities of bacillimide and bacillapyrrole from a marine-derived Streptomyces bacillaris strain. The research is very interesting that bacillimide inhibits the activity of Candida albicans isocitrate lyase. And the manuscript also determined that bacillimide inhibits the transcriptional level of icl in C. albicans under C2-carbon-24 by using growth phenotype method.

There are some questions should be addressed.

  1. Page 1, line 37: in the host environment, [3,4]. Delete ‘,’
  2. Page 1, line 37: been considered. To be ‘been considered as’
  3. Page 2, line 53: GenBank accession number: MK402083.1. GenBank accession number is base sequence not a strain.
  4. The NMR data in table are not consistent with Figures S1 and S6. And from Figure S1, I can’t find the coupling constants. 13C for C-9 of compound 2 is not consistent with Figure S16.
  5. Page 4, line 117: An additional correlation with H2-4. This signal should be from H2-4 to C-4.
  6. Page 4, line 133: reprocess the NOESY spectrum, I didn’t find peaks for H-7/H3-9, 6-OH/H-7.
  7. Page 5, line 154: The relative configurations are not solid, so the calculate ECD is not reliable.
  8. Page 6, line 162: m/z should be italic.
  9. Page 6, line 183-184: H3-10/C-8 and C-9, there is no peak from H3-10 to C-8 in Figure S19.

Author Response

[Response to Reviewer 1’s comments]

Thank you very much for your careful and valuable review of our manuscript. We made all revisions and corrections as far as we could. I hope this is the right answer for your request. What follows is our response to reviewer 1’s critique with the explanation of the changes implemented in the paper and a rebuttal when appropriate.

Comment 1:

Page 1, line 37: in the host environment, [3,4]. Delete ‘,’

Answer)

We appreciate the reviewer’s kind comments. ‘,’ was deleted in the revised version (Page 1, line 38).

Comment 2:

Page 1, line 39: been considered. To be ‘been considered as’

Answer)

According to the reviewer’s comments, ‘been considered’ (Page 1, line 40) was changed to ‘been considered as’.

Comment 3:

Page 2, line 53: GenBank accession number: MK402083.1. GenBank accession number is base sequence not a strain.

Answer)

We appreciate the reviewer’s comments. According to the reviewer’s comments, ‘(GenBank accession number MK402083.1)’ was deleted in the Introduction (Page 2, line 54) and described in the 3.2. Taxonomic Identification and Fermentation with reference in the revised version.

Comment 4:

The NMR data in table are not consistent with Figures S1 and S6. And from Figure S1, I can’t find the coupling constants. 13C for C-9 of compound 2 is not consistent with Figure S16.

Answer)

We appreciate the reviewer’s comments. Following the reviewer’s suggestion, the shifted 0.1 ppm of proton chemical shifts and carbon chemical shifts were corrected for Table 1, page 3.. Carbon chemical shift for C-9 of compound 2 was also changed from 53.8 to 53.6 ppm in Table 1, page 3. The expanded 1H NMR indicating coupling constants of the compounds 1 and 2 were attached to the supplementary materials (Figure S19 and Figure S20).

Comment 5:

Page 4, line 117: An additional correlation with H2-4. This signal should be from H2-4 to C-4.

Answer)

Although we appreciate the reviewer’s valuable comment, since we described the correlation from H2-4 to carbonyl carbon, not C-4 but C-2 is correct (Page 4, line 118).

Comment 6:

Page 4, line 133: reprocess the NOESY spectrum, I didn’t find peaks for H-7/H3-9, 6-OH/H-7.

Answer)

We apologize our mistake. Figure S6 was replaced with the correct NOESY spectrum which includes cross peaks for H-7/H3-9 and 6-OH/H-7.

Comment 7:

Page 5, line 154: The relative configurations are not solid, so the calculate ECD is not reliable.

Answer)

Although we appreciate the reviewer’s comment, it is hard to be agreeable. Since H-7, H3-9 and 6-OH showed NOESY correlations with spatially nearby protons (please see the revised NOESY spectrum), there would be no doubt that they are facing the same side, proposing the relative correlations. Furthermore, our NOESY-based relative configurations were identical to those of known compounds based on X-ray analysis.  

Comment 8:

Page 6, line 162: m/z should be italic.

Answer)

m/z was changed to italic in page 6, line 164.

Comment 9:

Page 6, line 183-184: H3-10/C-8 and C-9, there is no peak from H3-10 to C-8 in Figure S17.

Answer)

A peak from H3-10 to C-8 was not shown due to the partial overlapping. Figure S17 is replaced with an expanded spectrum (Figure S20).

Reviewer 2 Report

Table 1:

The coupling constants for CH2-4 and CH2-5 in compound 1 as well as the coupling constants for CH2-8 and CH2-9 in compound 2 should be adjusted.

Figure 2: The arrows on the structure of 1 and 2 should be simplified. Crowded arrows should be simplified. Arrows from adjacent carbons/hydrogens showing 1H-1H COSY correlations should be omitted

Supplementary Materials:

Expansion of the 1H signals of CH2-4 and CH2-5 in compound 1 as well as expansion of the 1H signals of CH2-8 and CH2-9 should be provided in the supplementary materials,

Author Response

[Response to Reviewer 2’s comments]

Thank you very much for your careful and valuable review of our manuscript. We made all revisions and corrections as far as we could. I hope this is the right answer for your request. What follows is our response to reviewer 2’s critique with the explanation of the changes implemented in the paper and a rebuttal when appropriate. 

Comment 1:

Table 1. The coupling constants for CH2-4 and CH2-5 in compound 1 as well as the coupling constants for CH2-8 and CH2-9 in compound 2 should be adjusted.

Answer)

We appreciate the reviewer’s comments. The expanded 1H NMR spectrum indicating coupling constants of the compounds 1 and 2 were attached to the revised supplementary materials (Figure S19 and Figure S20).

Comment 2:

Figure 2: The arrows on the structure of 1 and 2 should be simplified. Crowded arrows should be simplified. Arrows from adjacent carbons/hydrogens showing 1H-1H COSY correlations should be omitted.

Answer)

We appreciate for the reviewer’s comments. Accordingly, the arrows on the structure of 1 and 2 were simplified to remove those of the COSY-correlated signals in Figure 2.

Comment 3:

Supplementary Materials:

Expansion of the 1H signals of CH2-4 and CH2-5 in compound 1 as well as expansion of the 1H signals of CH2-8 and CH2-9 should be provided in the supplementary materials.

Answer)

We appreciate for the reviewer’s comments. The expanded 1H NMR spectrum indicating the coupling constants of the compound 1 was attached to the supplementary materials (Figure S19 and Figure S20).

Reviewer 3 Report

This paper reports on the isolation and characterization of two nitrogenous metabolites from Streptomyces bacillaris. Also, the inhibitory effects of the isolated compounds against isocitrate lyase enzyme is studied.

The assignment of the structure of compounds is based on spectroscopic analysis and mass spectrometry and it is adequate. The absolute stereochemistry of 1 was determined by the ECD-based computational method.

Corrections to be made:

Line 183 and figure 2: HMBC correlation H3-10/C-8 is unlikely to be observed since Methyl-10 and Methylene-8 are separated by four bonds. Also, I do not see the correlation in the supporting material information. I suggest to delete it from the text and figure 2.

On table 2 authors show the inhibitory activity against ICL enzyme of the isolated compounds and also of the known inhibitor 3-nitro-propionate. From the comparison of those values, I think the activities of 1 and 2 should not be described as “significant” for compound 1 and moderate for compound 2.

Author Response

[Response to Reviewer 3’s comments]

Thank you very much for your careful and valuable review of our manuscript. We made all revisions and corrections as far as we could. I hope this is the right answer for your request. What follows is our response to reviewer 3’s critique with the explanation of the changes implemented in the paper and a rebuttal when appropriate.

Comment 1:

Line 183 and figure 2: HMBC correlation H3-10/C-8 is unlikely to be observed since Methyl-10 and Methylene-8 are separated by four bonds. Also, I do not see the correlation in the supporting material information. I suggest to delete it from the text and figure 2.

Answer)

We appreciate for the reviewer’s comments. As the reviewer said, four bonds correlations are not common; however, we frequently find these correlations with favorable angles. Although the correlation was not strong, it was clearly shown on HMBC data. A peak from H3-10 to C-8 was not shown due to overlap. Figure S17 is replaced with an expanded spectrum (Figure S19 and Figure S20) instead of deleting this correlation from the text and Figure 2.

Comment 2:

On table 2 authors show the inhibitory activity against ICL enzyme of the isolated compounds and also of the known inhibitor 3-nitro-propionate. From the comparison of those values, I think the activities of 1 and 2 should not be described as “significant” for compound 1 and moderate for compound 2.

Answer)

We appreciate for the reviewer’s comments. According to reviewer’s comment, we carefully checked and revised our manuscript. The ICL enzyme inhibitory activities for compounds 1 and 2 were corrected for moderate and weak, respectively, in the Text and Abstract in the revised version.